# The Ophthalmic Manifestations of Down Syndrome

**DOI:** 10.3390/children10020341

**Published:** 2023-02-09

**Authors:** Emily Sun, Courtney L. Kraus

**Affiliations:** Wilmer Eye Institute, Johns Hopkins University, Baltimore, MD 21218, USA

**Keywords:** Down Syndrome, Trisomy 21, strabismus, amblyopia, nasolacrimal duct obstruction

## Abstract

Down Syndrome is one of the most common chromosomal conditions in the world, affecting an estimated 1:400–1:500 births. It is a multisystem genetic disorder but has a wide range of ophthalmic findings. These include strabismus, amblyopia, accommodation defects, refractive error, eyelid abnormalities, nasolacrimal duct obstruction, nystagmus, keratoconus, cataracts, retinal abnormalities, optic nerve abnormalities, and glaucoma. These ophthalmic conditions are more prevalent in children with Down Syndrome than the general pediatric population, and without exception, early identification with thoughtful screening in this patient population can drastically improve prognosis and/or quality of life.

## 1. Introduction

Down Syndrome is one of the most common chromosomal conditions in the world, affecting an estimated 1 in 400–1500 live births [1]. It is a multisystem genetic disorder, resulting from an extra copy of chromosome 21, either as a complete separate chromosome (most common), as a partial or full translocation, or existing in some but not all cells (mosaic form). This excess genetic material disrupts normal development, leading to characteristic physical features and developmental differences.

Down Syndrome manifests with a wide range of ophthalmic findings. These include strabismus, amblyopia, accommodation defects, refractive error, eyelid abnormalities, nasolacrimal duct obstruction, nystagmus, keratoconus, cataracts, retinal abnormalities, optic nerve abnormalities, and glaucoma. These ophthalmic conditions are more prevalent in children with Down Syndrome than the general pediatric population, and without exception, early identification with thoughtful screening in this patient population can drastically improve prognosis and/or quality of life. Therefore, it is important for physicians to be familiar with the ophthalmic manifestations in children with Down Syndrome in order to improve patient care and visual outcomes in this population. 

The purpose of this review is thus to describe the ophthalmic manifestations of children with Down Syndrome, discussing diagnosis, treatment, and management considerations. In doing so, we hope to help physicians in achieving the optimal visual outcomes for children with Down Syndrome. 

## 2. Strabismus

Strabismus has been shown to occur at higher rates in children with Down Syndrome compared to the normal population and has even been reported by some studies to be the most common ocular disorder within this population [2]. The incidence of strabismus in pediatric patients with Down Syndrome ranges from 9.5–57% [3,4,5,6,7,8,9]. 

In children with Down Syndrome, esotropia is the most common form of misalignment and is typically acquired between three and six years of age, as opposed to infantile onset [10]. This is later than in normally developing children, who typically develop acquired esotropia around the age of two [10]. The later onset of esotropia would suggest better binocular potential than those children with congenital esotropia if detected and treated appropriately. However, binocular vision in children with Down Syndrome has been reported to be subnormal [11]. 

The mechanism underlying the development of strabismus in children with Down Syndrome remains unclear. Strabismus has been found to be associated with hyperopia and accommodative insufficiency in patients with Down Syndrome [12]. It is proposed that this increased accommodative effort combined with dysfunction in fusional capacity and convergence predisposes children with Down Syndrome to develop strabismus [13]. 

Without treatment, strabismus can result in amblyopia and impair binocular visual development [14]. As such, early diagnosis is important. Strabismus in children with Down Syndrome is typically confirmed by a pediatric ophthalmologist performing a sensorimotor examination, which may be more challenging in this patient population. 

Pseudo strabismus, a condition where one or both eyes appear misaligned, but are actually straight, is a relatively common finding in patients with Down Syndrome. This is largely due to the facial structure associated with Down Syndrome. The broad, flat nasal bridge and prominent epicanthal folds create an “optical illusion” that the eyes are crossed [15]. Alternate cover testing is the “gold-standard” for distinguishing between pseudo strabismus and true strabismus and is performed as part of a comprehensive assessment by a pediatric ophthalmologist. However, for the pediatrician and situations where patient participation prevents more involved testing, the Hirschberg test, where a penlight or flashlight is projected towards the child and the light reflex is assessed, can be used. A light reflex located in the same location in each eye suggests aligned eyes (orthophoria), a temporally displaced light reflex reflects the inward movement of one eye and suggests true esotropia (Figure 1). 

Management of strabismus in patients with Down Syndrome may consist of hyperopic glasses with or without bifocals, extraocular muscle surgery, or both. Studies have supported the use of bifocals in children with Down Syndrome, reporting a reduction in the manifest angle of strabismus that persisted at a one-year follow-up [10]. For those whose ophthalmic course suggests surgery as the best option, surgical dosage for strabismus surgery does not need to be adjusted [16]. 

However, reports of an extreme postoperative inflammatory reaction have been linked more frequently to children with Down Syndrome following uncomplicated strabismus surgery. 

## 3. Amblyopia

Amblyopia has an increased incidence in children with Down Syndrome compared to the normal population [8,17], with rates reportedly ranging from 3–36.4% [6,8]. Larger variability in the prevalence of amblyopia has been suggested to be due to difficulties in assessing visual acuity in this population, which may lead to underdiagnosis [18]. 

Common causes of amblyopia in children with Down Syndrome include strabismus (53% of cases), 33% due to anisometropia, and 14% due to a strabismus and refractive error mixed etiology vs other (deprivation). Studies have also suggested that nasolacrimal ductal obstruction (NLDO) may lead to the development of amblyopia [19]. 

Treatment of amblyopia includes glasses and patching. Atropine drops have been proposed as an alternative treatment, particularly in children unable to tolerate patching. However, some caution may need to be exercised as few studies suggest that use of atropine may be contraindicated in patients with cardiac defects (such as those associated with Down Syndrome) due to increased sensitivity [20]. Surgical treatments may also be considered as a last resort for patients who are unable to tolerate the above therapies. 

## 4. Accommodation Defects

Children with Down Syndrome have been found to have reduced accommodative abilities [21,22]. This reportedly has an incidence ranging from 26–92% [3,23]. Haugen et al. found that 55% of children with Down Syndrome had reduced amplitude of accommodation, differing significantly from studies examining this ability in children without Down Syndrome [24]. Similarly, Cregg et al. found a substantial amount of underaccommodation in their study of children with Down Syndrome, and this underaccommodation was independent of refractive error [25].

The mainstay of treatment for accommodative defects is bifocals. Bifocals have been found to improve visual acuity and offset the accommodative insufficiency compared to single-vision lenses [12,26]. Additionally, among children with Down Syndrome, better compliance has been reported with bifocals compared to single-vision lenses [26,27]. In a follow-up study, children with Down Syndrome who were prescribed bifocals even had faster and improved performance on literacy skills compared to those prescribed single-vision lenses [28].

The authors generally recommend testing of accommodative amplitudes by the pediatric ophthalmologist caring for children with Down Syndrome. Especially in the setting of even mild hyperopia, use of bifocals is recommended. However, consideration of the individual patient including tolerance of glasses, presence of esotropia, mobility issues, among other issues, may encourage or discourage prescribing. 

## 5. Refractive Error

Refractive error is one of the most common ophthalmic findings of Down Syndrome. Hyperopia or myopia can be found, as well as anisometropia and astigmatism. Several studies have reported an incidence of refractive error of 80% or higher [29,30,31], which is significantly higher when compared to the general population [3,30]. There is conflicting evidence in the literature about whether hyperopia or myopia is more common in children with Down Syndrome. While some studies have found hyperopia to be the most common refractive error [6,27,30,31,32,33,34,35], other studies have found myopia to be more common [4,36,37]. 

Refractive error in children with Down Syndrome has been proposed to be linked to a failure in emmetropization, which is the natural decrease in refractive error that occurs during the first few years of life. Woodhouse et al. studied infants and children with Down Syndrome and examined the prevalence of refractive error in various age groups. While the distributions were not significantly different between infants with Down Syndrome and the control group, by preschool and primary school ages, the distributions were significantly wider, leading to a theory on a defect in emmetropization [38]. Haugen et al. similarly found that children with Down Syndrome were less likely to show a reduction in refractive error over time compared to the normal population [24]. This finding is supported by other studies, which have also found that refractive errors increase with age in children with Down Syndrome [31,32,39]. 

Refractive error in children with Down Syndrome is linked to strabismus [39], amblyopia, and impaired accommodative responses [23]. Prescribing guidelines are not different from children without Down Syndrome, with the exception of bifocals for potential accommodative issues [39]. 

## 6. Eyelid Abnormalities

Eyelid abnormalities are a common finding in children with Down Syndrome. Upward slant of the eyelids has been found to have an incidence between 82–100% among children with Down Syndrome, and also may be more prevalent in Asians [36]. Epicanthal folds are another common eyelid finding, and has been found to occur in 61–100% of children with Down Syndrome [35]. These abnormal eyelid morphologies may be linked to blepharitis and conjunctivitis, which have been found to have increased incidences in children with Down Syndrome compared to controls [3].

## 7. Nasolacrimal Duct Obstruction

Nasolacrimal duct obstruction (NLDO) is a problem with the drainage of the lacrimal system. This is typically due to a blockage at the valve of Hasner, but may be the result of other anatomical abnormalities such as canalicular stenosis [19,40,41]. Children with Down Syndrome have been found to have higher rates of incidence of NLDO compared to age-matched peers, with rates ranging from 3–35% [4,5]. The reason for this increased incidence is likely multifactorial, as patients with Down Syndrome often have unique facial morphologies that can cause NLDO. Some studies have reported that bilateral NLDO seems to be more common in children with Down Syndrome, as opposed to unilateral NLDO [19].

NLDO presents with persistent epiphora, tearing, and intermittent mucopurulent discharge involving one or both eyes. Pressure over the lacrimal sac typically induces the reflux of mucus. Abdu et al. found in their study of 175 Saudi children with Down Syndrome that the median age of diagnosis of NLDO was at one year, and tearing was the main presenting symptom [19]. Coats et al. also suggested that the dye-disappearance test can be used to confirm a diagnosis of NLDO, though acknowledged that this test may be difficult in uncooperative patients [40].

NLDO can be medically or surgically managed. While most cases of NLDO spontaneously resolve within the first year of life [40], this is less common among children with Down Syndrome, and with increasing age [19]. For cases that fail to resolve, most are surgically managed with high rates of success (~60–80%) [40,42]. Surgical treatment can consist of probing, balloon catheter dilation, or mono/bicanalicular stents. There is mixed evidence in the literature about the best technique and the ideal timing for these surgical interventions. Some studies suggest that early probing may prevent fibrosis in obstructed sites [19], while others suggest that balloon catheter dilation may have higher rates of success [43]. 

## 8. Nystagmus

Nystagmus is a common finding in children with Down Syndrome. The incidence of nystagmus has been reported to be as high as 33.3% [3,4,44]. In children with Down Syndrome, nystagmus has also been found to be associated with other ocular abnormalities, including congenital cataract [45], esotropia [7,44,46], accommodation defects [11], myopia [47], and other refractive errors [35,48].

Several types of nystagmus have been reported among children with Down Syndrome, though horizontal nystagmus has generally been found to be the most common [35,36,44,45,48,49]. In a study of 188 children conducted by Wagner et al., of the 56 children found to have nystagmus, 29 had fine rapid horizontal nystagmus, 14 had dissociated nystagmus which appeared pendular, and 9 had latent or manifest nystagmus [36]. Similarly, Postolache et al. found that in their study of 50 children with Down Syndrome that 15 had nystagmus, with all having horizontal nystagmus that was either latent-manifest or manifest [44]. Oladiwura et al. also described in their study that “manifest horizontal nystagmus” was the main descriptive term used in the medical records of 40 out of 48 children with a diagnosis of Down Syndrome and nystagmus [45]. The mechanism underlying nystagmus in children with Down Syndrome remains unknown, but is also likely multifactorial in nature. 

## 9. Keratoconus

Keratoconus is a progressive thinning and bulging of the cornea, which can result in irregular astigmatism and decreased visual acuity. It typically develops in adolescence or young adulthood. In the studies that specifically examined children with Down Syndrome, the incidence of keratoconus was reportedly between 0–32% [50]. In adults with Down Syndrome, the incidence is as high as 71.3% [51]. One Norwegian study even found that keratoconus is 30 times more prevalent in patients with Down Syndrome compared to the general population [52].

Even in the absence of the clinical manifestations of keratoconus, children with Down Syndrome have been shown to have various corneal abnormalities. Vincent et al. and Imbornoni et al. both found that children with Down Syndrome had a steeper corneal curvature compared to controls [53,54]. Other studies have also shown that children with Down Syndrome have decreased central corneal thickness [54,55,56,57]. 

The development of keratoconus is likely a combination of hereditary, environmental, and cellular factors. The corneal abnormalities associated with Down Syndrome have been hypothesized to partially predispose these children to develop keratoconus [53]. Additionally, an abnormality in collagen cross-linking has been proposed as a potential explanation for the genetic link to keratoconus. Specifically, the gene encoding alpha-1 chain of type VI collagen is located on chromosome 21, and has been a target for research [55]. Finally, eye rubbing is a risk factor for developing keratoconus, which may serve as another potential explanation for the increased incidence of keratoconus in patients with Down Syndrome [29,41,50,54].

Pediatricians may encounter reports of high astigmatism on in-office vision screeners. Early screening for keratoconus includes retinoscopy and assessment of the cornea curvature on slit lamp examination. Progression of astigmatism and steepening of the cornea suggest potential keratoconus. Treatment can range from glasses to surgical procedures including corneal transplants or cross-linking. 

## 10. Iris

Brushfield spots are benign areas of iris stromal connective tissue hyperplasia, appearing as speckled spots on the iris. Brushfield spots have an increased incidence in children with Down Syndrome, ranging from 13–90% [35]. Studies have suggested that Brushfield spots may be more prevalent in children with lighter eye colors [35]. 

## 11. Cataracts

Congenital cataracts have a reported incidence of 1.4–50% among children with Down Syndrome [58]. Although some studies suggest that cataracts usually appear between the age of 12–15 [59], others report cataracts in younger children [35,48]. Congenital cataracts among children with Down Syndrome are 10-fold higher than in the general population [5], and one study found that 76% of children with Down Syndrome undergoing cataract surgery had congenital cataracts [32].

Delayed treatment of cataracts can result in amblyopia and low vision [4]. Additionally, one study found that children with Down Syndrome and cataracts have greater developmental quotients [60]. As such, the early diagnosis and treatment of cataracts in children with Down Syndrome is particularly important.

Not all cataracts require surgery, including anterior polar cataracts, which are common in this population. In cataracts that require surgical management, visual outcomes are thought to be improved [41]. However, the literature has mixed evidence regarding rates of post-operative complications. Some studies have shown that the rate of post-operative complications is comparable between children with Down Syndrome and the normal pediatric population [61,62]. However, other studies have found that the risk of post-operative complications is still high in frequency in children with Down Syndrome [63], and may range from 20–60% [64]. 

## 12. Retinal Abnormalities

Retinal abnormalities have also been found to have an increased incidence in children with Down Syndrome, with reports ranging from 1.7–40% [3,4]. These retinal abnormalities can present in a variety of ways, and include problems like retinal degeneration, chorioretinitis, preretinal hemorrhage, torturous vessels, increased retinal thickness and retinal detachment. 

One common retinal abnormality reported in children with Down Syndrome is abnormal retinal vasculature, with the optic disc appearing more crowded. These studies have found that there are more large vessels crossing the disc margin in children with Down Syndrome [65], with one study finding that 15% of children with Down Syndrome had eighteen or more retinal vessels crossing the optic nerve head margin [48].

One of the most dreaded retinal complications among children with Down Syndrome is retinal detachment. In patients with Down Syndrome, retinal detachment is often due to trauma. One study found that trauma was implicated in 55% of cases of patients with Down Syndrome [66]. While the anatomic success rate of retinal reattachment surgery in patients with Down Syndrome is comparable to the general population, this patient population is more likely to present with more chronic and complex retinal detachment [67]. 

There is some concern that retinal neoplasms may be slightly more common in patients with Down Syndrome, specifically involving the retinal pigment epithelium [68]. As regular dilated eye examinations are already recommended for patients with Down Syndrome, additional screening is unnecessary. 

## 13. Optic Nerve Abnormalities

Optic nerve abnormalities have also been found to be associated with Down Syndrome. One study conducted by Scheiner et al. found that in a retrospective review of 806 children with Down Syndrome, 14% had optic nerve abnormalities, with the most frequent abnormality being a myopic appearing nerve with peripapillary atrophy. Other optic nerve abnormalities they found included elevated optic nerve head, pigmentary changes, scleral crescent, optic nerve pallor, and non-myopic peripapillary atrophy [69]. Other studies have found a prevalence of optic nerve head drusen ranging from 6.8–8% in children with Down Syndrome [8,44]. Pseudotumor cerebri has also been reported to have a prevalence of 3.4% in children with Down Syndrome [70]. Glaucoma is a rarer finding in Down Syndrome, but can have devastating consequences of vision loss. The incidence of glaucoma among children with Down Syndrome has been reported to be between 0–6.7% [4].

## 14. Conclusions

Down Syndrome is associated with numerous ophthalmic conditions, which can result in poor vision and reduced quality of life in children with Down Syndrome. As such, it is crucial for clinicians caring for children with Down Syndrome to be aware of these ophthalmic manifestations, in order to detect and treat these conditions early. Knowledge of these conditions and their management considerations will allow clinicians to optimize vision and minimize ocular complications in these patients. 

## Figures and Tables

**Figure 1 children-10-00341-f001:**
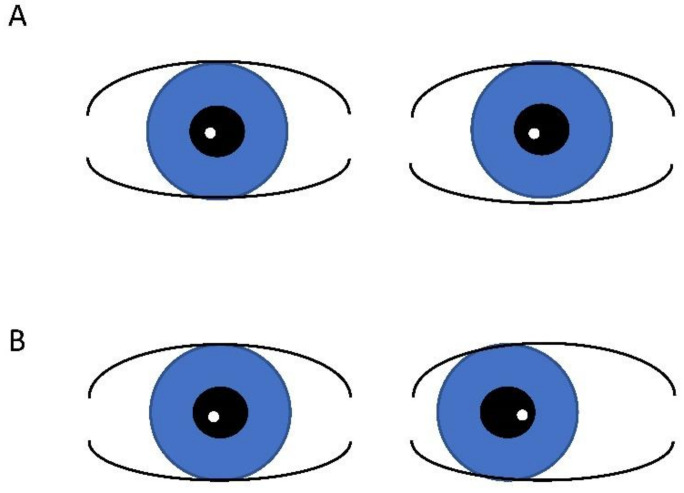
Hirschberg test demonstrating light reflex in same location in aligned (orthophoric) eyes (**A**) and temporal displacement of the light reflex in esotropia (**B**).

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
