# Peer review of "The Ophthalmic Manifestations of Down Syndrome"

_children, 2023, doi:10.3390/children10020341_

Round 1

Reviewer 1 Report

Dear Editor,

Firstly, I want to thank you for giving me this opportunity. I have carefully read and reviewed this “Review” type of paper: “A review of ocular findings in pediatric patients with Down Syndrome”

First of all, I would like to congratulate the authors for such an important review in this field.

The manuscript is written about to describe the ophthalmic manifestations of children with Down Syndrome, discussing diagnosis, treatment, and management considerations.

The research content is in accordance with the aim and scope of the journal. Text language is sufficient, fluent and understandable.

This is a fine work that I really enjoy reading although there are a few issues that the authors need to clarify. If the authors consider the suggestions I have mentioned below, the manuscript will be significantly improved.

Good luck and success.

1)     First of all, if this is not an ophthalmology journal [Children (ISSN 2227-9067)], readers will generally be in pediatrician and the like. For this reason, it would be useful to consider screening and treatment methods. While the authors stated their aims, they also stated that they would talk about treatment, and management. However, they generally only talked about diagnosis and general information. It would be good for them to reconsider this issue. Example: keratoconus diagnosis techniques. Keratoconus treatment? follow up? Check out all the main topics including keratoconus.

2)     You mentioned Accommodation Defects in amblyopia. My personal suggestion is that you present this as a separate topic (title). The biggest problem that is overlooked routinely in Down syndrome patients is the lack of “accommodation”.

3)     The point I would like to add: Should all kids with down syndrome use executive bifocal glasses that enhance close vision, beneficial for development up to school age and even to the end of primary school? Can you address this issue?

4)     In relation to the above comment and Figure 1, where should the executive bifocal line align? Can you specify? Will it be more effective if "near side" of bifocal contains within the pupil level in a little bit?

5)     Extraocular malignancy has a serious place in Down syndrome. What about ocular neoplasms?

6)     Retinal detachment and Down syndrome? would you like to talk about this relationship?

7)     Last but not least, it is known that patients with Down syndrome have serious mental retardation and intellectual disability. It is known that such patients are frequently exposed to trauma. Therefore, is there an ocular trauma and Down syndrome relationship?

Kind regards

Author Response

- line 60 the Authors advises that an alternate cover test can help distinguish between pseudostrabismus and true strabismus. This is true, but it is obvious procedure. If we want to check if strabismus is present, CT is one of the basic tests. You don't have to give it. If the authors planned to advices how to detect this disorder, it is worth mention about other useful tests for strabismus, such as the Hirshberg test or the Bruckner test, which can be performed even in patients who do not cooperate well. With CT, the patient must cooperate, and children with Down syndrome often have a problem with this (especially in young age)

Thank you for this comment, we agree, it is important to mention to the reader and outline for a potential primary pediatric provider. We created a figure (Figure 1) as well as added text to illustrate this point. 

- Strabismus is strongly related to the lack of binocular vision and stereopsis. Are there any raport about the level of binocularity with Down syndrome in the literature ?

Limited, but we addended to mention.   

It is worth mentioning the research with ERG and VEP in Down's syndrome. It would show the organization and function of the retinal nerve cells and the visual pathway from the retina to the brain.

We appreciate this suggestion, with all the other comments and additions we feel we may not have room to include this discussion, electing to omit only this addition from the reviewers' excellent comments bc the technical nature may be less relevant to pediatricians. 

- line 41-42, sentence "The incidence of strabismus among pediatric patients ranges from 9.5%-57%" is not clear. % refers to general pediatric population or population with Down syndrome?

addressed. Thank you

- citation, the numbers should be in brackets; actually they blend in with the text

Yes, thank you, this was also an error in the conversion of our document to journal format. 

- line 86 - Accommodation Defects should be bolded and became a new item 4, and then in sequence.

Thank you, for this identification. This was how we had originally submitted it- as its own category, must have been changed in formatting. We will ensure our update has it appropriately bolded.   

Reviewer 2 Report

The presented paper, based on the literature, have shown that vision problems in children with Down syndrome are much more common than in healthy children, which may affect their overall development. The authors have extensively analyzed the literature, selecting the most important articles in this field.

The work is clinically significant, demonstrating to other specialists the importance of early and regular eye checkups in children with Down syndrome.

Below are points and suggestions for changes:

- line 60 the Authors advises that an alternate cover test can help distinguish between pseudostrabismus and true strabismus. This is true, but it is obvious procedure. If we want to check if strabismus is present, CT is one of the basic tests. You don't have to give it. If the authors planned to advices how to detect this disorder, it is worth mention about other useful tests for strabismus, such as the Hirshberg test or the Bruckner test, which can be performed even in patients who do not cooperate well. With CT, the patient must cooperate, and children with Down syndrome often have a problem with this (especially in young age)

- Strabismus is strongly related to the lack of binocular vision and stereopsis. Are there any raport about the level of binocularity with Down syndrome in the literature ?

It is worth mentioning the research with ERG and VEP in Down's syndrome. It would show the organization and function of the retinal nerve cells and the visual pathway from the retina to the brain.

- line 41-42, sentence "The incidence of strabismus among pediatric patients ranges from 9.5%-57%" is not clear. % refers to general pediatric population or population with Down syndrome?

- citation, the numbers should be in brackets; actually they blend in with the text

- line 86 - Accommodation Defects should be bolded and became a new item 4, and then in sequence.

Author Response

1)     First of all, if this is not an ophthalmology journal [Children (ISSN 2227-9067)], readers will generally be in pediatrician and the like. For this reason, it would be useful to consider screening and treatment methods. While the authors stated their aims, they also stated that they would talk about treatment, and management. However, they generally only talked about diagnosis and general information. It would be good for them to reconsider this issue. Example: keratoconus diagnosis techniques. Keratoconus treatment? follow up? Check out all the main topics including keratoconus. Thank you for this very reasonable point, we have edited to address for this topic in our manuscript 

2)     You mentioned Accommodation Defects in amblyopia. My personal suggestion is that you present this as a separate topic (title). The biggest problem that is overlooked routinely in Down syndrome patients is the lack of “accommodation”. Yes, this was our original intent and it was altered in formatting. Addressed. 

3)     The point I would like to add: Should all kids with down syndrome use executive bifocal glasses that enhance close vision, beneficial for development up to school age and even to the end of primary school? Can you address this issue? I think it is difficult to say that all children with Down syndrome should use executive bifocal glasses, it is certainly the preference of many prescribing pediatric ophthalmologists. However, we want to avoid such an absolute statement as there are many nuances that the ped ophth may be considering and for the pediatric practitioner reading this review, we just hope to inform about the benefits that may exist for this technology. We hope we added sufficient detail to address your point. 

4)     In relation to the above comment and Figure 1, where should the executive bifocal line align? Can you specify? Will it be more effective if "near side" of bifocal contains within the pupil level in a little bit? For the interest of word constraints we will reserve comment on these fit issues as the prescribing MD/DO should hopefully be handing. But are open to a discussion for its need for this review. The authors personally recommend bifocals bisect pupils unless there are serious mobility concerns (for fear of altering downsize vision clarity). 

5)     Extraocular malignancy has a serious place in Down syndrome. What about ocular neoplasms? Discussed and included, thank you for pointing out this oversight.

6)     Retinal detachment and Down syndrome? would you like to talk about this relationship? This is included in the retinal section. 

7)     Last but not least, it is known that patients with Down syndrome have serious mental retardation and intellectual disability. It is known that such patients are frequently exposed to trauma. Therefore, is there an ocular trauma and Down syndrome relationship? See above, mostly in regards to retinal detachments and occasionally cataracts. 

Round 2

Reviewer 1 Report

Dear Editor,

The authors really seem to have done their best. 

Now the manuscript has become more convenient and improved.

I do not see any inconvenience to endorse.

Kind regards

Reviewer 2 Report

The Authors included additional information and a present form is ready for publishing.